# Mucopolysaccharidosis Type I: A Review of the Natural History and Molecular Pathology

**DOI:** 10.3390/cells9081838

**Published:** 2020-08-05

**Authors:** Christiane S. Hampe, Julie B. Eisengart, Troy C. Lund, Paul J. Orchard, Monika Swietlicka, Jacob Wesley, R. Scott McIvor

**Affiliations:** 1Immusoft Corp, Seattle, WA 98103, USA; monika.swietlicka@immusoft.com (M.S.); jake.wesley@immusoft.com (J.W.); 2Department of Pediatrics, University of Minnesota, Minneapolis, MN 55455, USA; eisen139@umn.edu (J.B.E.); lundx072@umn.edu (T.C.L.); orcha001@umn.edu (P.J.O.); 3Immusoft Corp, Minneapolis, MN 55413, USA; r.scott.mcivor@immusoft.com or; 4Department of Genetics, Cell Biology and Development and Center for Genome Engineering, University of Minnesota, Minneapolis, MN 55413, USA

**Keywords:** mucopolysaccharidosis type I, animal models, α-L-iduronidase

## Abstract

Mucopolysaccharidosis type I (MPS I) is a rare autosomal recessive inherited disease, caused by deficiency of the enzyme α-L-iduronidase, resulting in accumulation of the glycosaminoglycans (GAGs) dermatan and heparan sulfate in organs and tissues. If untreated, patients with the severe phenotype die within the first decade of life. Early diagnosis is crucial to prevent the development of fatal disease manifestations, prominently cardiac and respiratory disease, as well as cognitive impairment. However, the initial symptoms are nonspecific and impede early diagnosis. This review discusses common phenotypic manifestations in the order in which they develop. Similarities and differences in the three animal models for MPS I are highlighted. Earliest symptoms, which present during the first 6 months of life, include hernias, coarse facial features, recurrent rhinitis and/or upper airway obstructions in the absence of infection, and thoracolumbar kyphosis. During the next 6 months, loss of hearing, corneal clouding, and further musculoskeletal dysplasias develop. Finally, late manifestations including lower airway obstructions and cognitive decline emerge. Cardiac symptoms are common in MPS I and can develop in infancy. The underlying pathogenesis is in the intra- and extracellular accumulation of partially degraded GAGs and infiltration of cells with enlarged lysosomes causing tissue expansion and bone deformities. These interfere with the proper arrangement of collagen fibrils, disrupt nerve fibers, and cause devastating secondary pathophysiological cascades including inflammation, oxidative stress, and other disruptions to intracellular and extracellular homeostasis. A greater understanding of the natural history of MPS I will allow early diagnosis and timely management of the disease facilitating better treatment outcomes.

## 1. Introduction

The mucopolysaccharidoses (MPS) are rare genetic disorders that affect lysosomal degradation of mucopolysaccharides (glycosaminoglycans (GAGs)) [1]. GAGs are a major component of the extracellular matrix (ECM), where they promote cell-to-cell and cell-to-ECM adhesion. MPS patients are deficient in one of the lysosomal enzymes involved in the stepwise degradation of GAGs, leading to accumulation of partially degraded GAGs in lysosomes and the ECM. Mucopolysaccharidosis type 1 (MPS I) is caused by lack, or relative deficiency, of the enzyme α-L-iduronidase (IDUA), leading to the accumulation of dermatan and heparan sulfates [1]. Three MPS I subtypes have been classified that differ in the severity of disease, ranging from mild (Scheie syndrome) to moderate (Hurler–Scheie) to severe (Hurler syndrome or MPS-IH). Severe disease, i.e., MPS-IH, is caused by absence or extremely low levels of IDUA, associated with genotypes such as deletions and nonsense mutations. It is now appreciated that MPS I exists as a spectrum of disorders from an attenuated version to severe with many phenotypes in between.

At birth, the afflicted neonate appears healthy; and symptoms that develop during the first 6 months of life are vague and thus, diagnosis is often delayed [2,3]. Early diagnosis and treatment are, however, crucial to prevent development of severe manifestations. The few studies conducted in fetuses with MPS I revealed greatly elevated GAG levels in the liver [4], whereas the brain showed only mildly elevated levels [5,6]. These findings suggest that the initially affected organ is the liver, while other organs, including the brain, are affected later and their dysfunction is likely the result of additional secondary insults from the pathological cascades of GAG accumulation [1,7].

To our knowledge, no prenatal studies have been conducted in animal models for MPS I. However, prenatal MPS VII mice show delayed development of ossification centers in the tibia, suggesting an early onset of dysostosis multiplex in this form of MPS [8].

In the following sections, we will discuss common phenotypic manifestations of the disease, in the order in which they typically develop. Key manifestations are described in detail. Cardiac disease is discussed as a separate manifestation, because age at onset of cardiac manifestations is less well defined and can range from infancy to older individuals.

## 2. Symptoms during the First 6 Months of Life

Early symptoms appearing during the first 6 months of life include recurrent rhinitis, coarse facial features, hernias, thoracolumbar kyphosis, and hepatosplenomegaly [2,9,10,11,12,13] (Table 1).

### 2.1. Facial Features

Parents’ early observation often includes their child’s unusual appearance, including hirsutism, enlarged head, and characteristic facial features [14]: a broad nose and flared nostrils, prominent supraorbital ridges, large rounded cheeks, thick lips, and an enlarged protruding tongue. These characteristic features are caused by GAG deposits in the soft tissues and bones. Often the sagittal suture fuses prematurely (scaphocephaly). The skin may be thickened and inelastic [14]. Similar facial abnormalities are observed in the animal models of MPS I. MPS I cats have facial dysmorphia, with a thick neck, large head, frontal bossing, depressed nasal bridge, and small ears [15]. MPS I dogs show frontal bossing and broad, concave nasal shape, and thickened skin at the scruff of the neck [16,17]. The canine facial dysmorphisms develop early, around 3–6 months of age [16]. Craniofacial manifestations in MPS I mice develop as a flattened facial profile at 3–5 weeks of age, after which, the face becomes progressively broader [18].

### 2.2. Abdominal Hernias

Recurrent umbilical and inguinal hernias are common in MPS I patients and are attributed to an aberrant collagen state in the abdominal wall [19]. Umbilical hernias have also been described in the canine MPS I model [16].

### 2.3. Hepatosplenomegaly

Protuberance of the abdomen caused by progressive hepatosplenomegaly is common [18]. Although the organs may be vastly enlarged, storage of GAGs in the liver and spleen does not lead to organ dysfunction. Hepatosplenomegaly is noted also in MPS I mice with an onset at 8–10 weeks of age [18,20].

### 2.4. Respiratory and Pulmonary Manifestations

Recurrent rhinitis and upper airway obstructions without apparent infection are common early manifestations of MPS I (Figure 1). Rhinitis is likely exacerbated by narrow nasal passageways causing obstructions that limit normal mucosal drainage. Musculoskeletal deformities such as nasal dysmorphism, short neck, abnormal cervical vertebrae, and mandible predispose the patient for the development of upper airway obstructions. Moreover, extracellular and intracellular GAG deposits in the soft tissues of the mouth, nose, and throat result in enlargement of the tongue, adenoids, and tonsils [21,22]. Upper airway obstruction manifests in noisy breathing, snoring, and sleep apnea. This often results in infections of the respiratory tract and of the ear. An analysis of 615 Hurler patients in the MPS I registry revealed that 85% of patients suffered from airway-related symptoms, including sleep disturbances, enlarged tonsils, and macroglossia [23]. However, other underlying causes for sleep apnea in MPS I must be considered, including neurological abnormalities, such as hydrocephalus [24].

Initially, respiratory symptoms are more pronounced in the upper airways, but as the disease progresses, tracheobronchial manifestations can emerge, which in severe cases are frequently the cause of death. GAG deposits cause narrowing of the tracheal lumen and deformities of the trachea itself [25,26]. Alterations in collagen fibers can further weaken the trachea and result in tracheomalacia and bronchomalacia. In combination with bronchial and tracheal stenosis, this may lead to a compression of the airways [27,28]. Finally, skeletal deformities of the thoracic cage predispose MPS I patients for restrictive lung disease [29]. Milder clinical manifestations include dyspnea, difficulty clearing secretions, cough, wheezing, and recurrent bronchitis or pneumonia [22]. Despite the obvious importance of respiratory issues, these are rarely investigated in the animal models of MPS I. Schuh et al. used plethysmography to establish that lung function is significantly compromised in MPS I mice, however, no histopathological analysis was conducted [30].

### 2.5. Diagnostic Tests of Respiratory Function

The early identification of respiratory problems is important to maximize treatment outcome. Initial imaging including X-ray, computed tomography, and rigid bronchoscopy can help identify anatomical airway obstruction or skeletal changes involved. Other evaluations of the upper airway anatomy can be made with the help of the Mallampati classification system (oropharynx) and Brodsky grading scale (tonsils) [31]. Pulmonary function is commonly tested by spirometry. Typical measurements consist of expiratory forced vital capacity (FVC), forced expiratory volume in one second (FEV1), and the ratio of these two measurements, FEV1/FVC (Tiffeneau Index). Obstructive disease is indicated by a reduction in FEV1 (expressed as FEV1/FVC < 70%), whereas restrictive dysfunction is indicated by low values for FVC and FEV1 and a normal or increased FEV1/FVC ratio [22]. Interpretation of a spirogram relies on age-, sex-, and height-dependent reference values [32], which is complicated in MPS I patients because of their additional disease manifestations such as growth impairment and skeletal abnormalities. Importantly, spirometry requires the patient’s cooperation, and performance can be affected by age and/or limited cognitive function [33]. Still, spirometry remains a meaningful clinical tool and is used for screening, to determine longitudinal disease progression and response to therapy. Many patients with MPS I suffer from sleep-disordered breathing (SDB), including obstructive sleep apnea (OSA) and/or sustained hypoventilation. Therefore, sleep studies/polysomnography are recommended for all MPS I patients [29].

## 3. Symptoms Developing after 6 Months of Age

After 6 months of age, other more specific symptoms begin to emerge, such as impaired hearing and vision, and more pronounced musculoskeletal defects, followed by psychomotor retardation and cardiovascular defects (Table 1) [2,9,12].

### 3.1. Hearing Loss

Hearing loss in patients with MPS IH is common and presents often during the first year of life. Both conductive and sensorineural components are involved. The conductive hearing loss may be caused by frequent middle ear infections, middle ear mucus, Eustachian tube dysfunction, and thickened middle ear mucosa by extracellular and intracellular GAG deposits. Recurrent otitis media is a universal problem with the disease and can be a presenting symptom [21,34].

Understanding of the pathophysiology of the sensorineural component of hearing loss is based on several autopsy reports of MPS IH patients and studies in animal models. Autopsies revealed GAG deposits in cells of both the middle and inner ear (Table 2) (Figure 2). The middle ear presents with thickened tympanic membranes [35], which may be perforated [36], and excessive mesenchymal tissue can fill the middle ear cavity [35,37,38]. Both phenomena may contribute to hearing loss. The tympanic mesenchyme tissue usually resolves before birth and residual mesenchyme in the middle ear has been correlated with otitis media [39]. A thick GAG-positive mucosal lining also covers the ossicles, which may be causal to chronic secretory otitis media [36]. Both the mastoid process and ossicles show cells laden with undegraded GAGs [36,37].

The inner ear in MPS IH patients shows extracellular GAG deposits and GAG accumulation in cells in the majority of tissues [36]. Progression to degeneration of the organ of Corti has been reported in some studies [36,37], as was loss of cochlear hair cells [35]. Moreover, Reissner’s membrane has been found adhered to the organ of Corti (Figure 2) [36,37,40]. The presence of GAG deposits in the stria vascularis and fibrocytes in the spiral ligament has been reported in some cases [35,37], whereas other studies found no such abnormalities [36]. Autopsy results of two MPS IH patients found damage to the cochlear nerve caused by disruption of nerve fibers by infiltrating GAG-laden cells [36], and Friedmann speculated that the loss of nerve cells from the spiral ganglion may be secondary to cochlear nerve damage [36].

In MPS I mice, otitis media presents at 2 months of age [41] (Table 2). At that time, the mice experience mild hearing loss, and lysosomal storage in a number of different cell types of the inner ear can be observed. At 6 months, the hearing loss is profound and lysosomal storage is markedly increased, particularly in the fibrocytes of the cochlear wall [41]. Notably, at this time, cochlear hair cells remain intact and show no sign of lysosomal storage. However, at 1 year of age, the animals are deaf and cochlear hair cells are absent [41,42]. The profound loss of hearing at 6 months in the presence of normal hair cells suggests that loss of these cells may be secondary to lysosomal storage and degeneration of other cells. In support of this scenario, Schachern postulated that degeneration of the cochlear fibrocytes is involved in the hair cell loss. These fibrocytes play a significant role in maintaining the ion homeostasis of the cochlea and their degeneration has been reported to precede loss of cochlear hair cells in other mouse models [43].

MPS I dogs show infiltration of GAG-laden macrophages causing thickened tympanic membranes and thickened tissue encasing the ossicles (Table 2). The ossicles themselves show enlarged osteocytes and chondrocytes [44]. However, otitis media is not common in the animals. The dogs’ inner ear shows GAG accumulation in different cell types. Most significantly, GAG-deposits in the cochlear nerve causes the disruption of nerve fibers in MPS I dogs, seen as early as 1.6 months. In stark difference to the MPS I mouse model, no hair cell loss was observed, despite enlarged fibroblasts present in the spiral ligament, further supporting the conclusion that loss of hair cells is not necessary for MPS I-associated hearing loss.

#### Diagnostics for Auditory Manifestations

Because of the high frequency of hearing loss, its early onset, and progressive nature, auditory tests should be performed regularly in MPS I patients. These tests consist of audiometry, including bone conductive tests [45], and comprehensive evaluations of the involvement of different parts of the ear [46,47] (Table 3). Assessment of auditory function can be complicated by the age of the patient and/or cognitive function and/or because of the presence of ventilation tubes inserted in the tympanic membrane.

Neuroimaging is rarely included in the audiology exam, but results of magnetic resonance imaging (MRI) of the ear can be useful to confirm outcomes of the hearing tests [47].

A recent comprehensive study of patients with MPS including 10 MPS I participants used different audiometry procedures, depending on age and cognitive function [48]. Hearing in children aged 6 months to 2 years was tested by visual reinforcement audiometry (VRA), whereas children aged 2–6 years underwent conditional tonal play audiometry and individuals aged 7 and older were tested through conventional tonal audiometry. The cognitive function of each subject was considered in the choice of hearing test. The findings showed predominantly conductive and mixed hearing loss in MPS I, MPS II, and MPS VI patients.

The middle ear can be tested by tympanometry to evaluate compliance of the middle ear, acoustic reflex testing (ART) to differentiate between conductive hearing loss and mild/moderate sensorineural hearing loss and static acoustic impedance to determine blockage or eardrum perforation. In young children and/or individuals lacking in behavioral assessment, responses to otoacoustic emissions (OAE) and brainstem auditory evoked potentials (BAEP) can be employed to estimate or confirm auditory thresholds and further investigate sensorineural defects of the inner ear.

### 3.2. Ocular Manifestations

Ocular manifestations of MPS I are caused by GAG deposits in the majority of ocular tissues and include corneal clouding, ocular hypertension/glaucoma, retinal degeneration, optic nerve swelling/atrophy [49], refractive errors, and ocular motility abnormalities [50] (Figure 3) (Table 4).

#### 3.2.1. Corneal Clouding

The cornea consists of a surface epithelium, a subepithelial, acellular Bowman’s layer, the stroma, Descemet’s membrane, and the endothelium. The stroma makes up 90% of the corneal thickness and contains collagen, GAGs, proteoglycans, and keratocytes. The stroma-embedded keratocytes synthesize and degrade both collagens and GAGs. Collagen is arranged in fibrils of remarkably uniform diameter and the fibrils are arranged in parallel in lamellae. This highly organized arrangement is essential for the transparency of the cornea and is regulated by proteoglycans attached to collagen fibrils [51]. Corneal clouding is caused by aberrant GAG deposits in stromal keratocytes and disruption of normal collagen alignment in the corneal stroma [49,52]. The usually highly uniform collagen fibrils show great variations in diameter, and the fibrils are spaced further apart [53]. Corneal clouding usually affects both eyes and progresses from photophobia in the early stages to a milky, ground glass appearance [54]. It is often an early, albeit mild, sign and occurs within the first year of life [2,55]. Corneal clouding increasingly affects visuality and can lead to blindness, especially if associated with other ocular manifestations.

Corneal clouding is mimicked particularly faithfully in the MPS I feline model [56]. MPS I dogs show corneal clouding at 6 months of age [16], with characteristic GAG accumulation and cellular enlargement of keratocytes and disorganized collagen fibrils [57], similar to that reported in human MPS I. In MPS I mice, accumulation of GAG in corneal stromal cells can be detected at 2 months of age. However, the fibril structure becomes affected only at 8 months of age [58].

#### 3.2.2. Optic Nerve Swelling

Optic nerve swelling has been described at a frequency of over 50% in Hurler patients [59], although later investigations could not confirm this high frequency [54,60]. Swelling of the optic nerve can be caused by raised intracranial pressure, secondary to GAG-mediated accumulation of cerebrospinal fluid in the brain. Other causes of optic nerve swelling are scleral thickening [61] or GAG accumulation within the nerve and the meningeal sheath [62,63].

#### 3.2.3. Retinopathy

Variable degrees of retinopathy have been reported in MPS I and generally manifest later in life [64,65,66]. GAG deposits within retinal pigment epithelial cells and in the photoreceptor matrix leads to progressive photoreceptor loss, retinal degeneration, and dysfunction [49,63,67]. Clinically, patients may first complain of sensitivity to light and night blindness, progressing to tunnel vision and eventually to central visual field loss [49].

In MPS I cats, GAG deposits were observed in the retinal pigment epithelium (RPE), although the cells were not enlarged. In contrast to human MPS I, no abnormalities were found in the retinal photoreceptors or ganglion cells of MPS I cats [56]. Retinal degeneration in MPS I mice occurs with no obvious GAG deposits in the retinal tissues [58]. The outer nuclear layer (ONL) of the retina showed progressive reduction in thickness due to loss of photoreceptor cells, which was significant at 6 months of age. The inner nuclear layer (INL) remained normal. Dogs with MPS I do not demonstrate GAG accumulation in the retinal cells or the anterior chamber structure, so these dogs do not show manifestations of glaucoma or optic nerve degeneration [68].

#### 3.2.4. Glaucoma

In the normal eye, aqueous humor flows from the posterior chamber through the pupil to the anterior chamber and exits through the trabecular meshwork. In glaucoma, the drainage of aqueous humor is impaired and pressure increases in the posterior chamber. In MPS I, drainage may be impaired by GAG accumulation in the trabecular meshwork (open-angle glaucoma) or by GAG deposits in the iris and cornea, which bring the lens and the iris into contact and eventually prevents drainage (closed-angle glaucoma). Increase in intraocular pressure (IOP) can eventually lead to optic nerve damage [69,70]. Although glaucoma and ocular hypertension have been described for MPS IH patients, it is relatively rare, affecting about 4% of patients [70,71].

#### 3.2.5. Diagnostics for Ocular Manifestations

Frequent (half-yearly to annual) assessments of ocular manifestations are recommended for patients with MPS I. Because of the variety of ocular manifestations, different tests are necessary to determine the patient’s individual condition (Table 5). Basic function tests should assess visual acuity, coordinated function of both eyes (binocular function), and ocular motility. Corneal clouding can be examined by a slit lamp exam or by Iris Camera [72]. Possible optic nerve damage can be assessed by the pupil’s reactions to light, however, photophobia can interfere with this eye exam. Other exams investigating optical nerve damage include visual field evaluation and fundus evaluation. Retinal degeneration can be assessed by color vision tests. Measurement of IOP for the determination of glaucoma can be falsely high due to interference of increased corneal thickness [63] or increased rigidity of the cornea [73], and different techniques can be chosen for the measurement of IOP in MPS I [74]. Additional eye exams can be conducted to investigate other aspects of MPS I-related ocular manifestations as outlined elsewhere [72,73,75].

## 4. Skeletal Disease and Joint Symptoms in MPS I

Skeletal disease in MPS I, collectively described as dysostosis multiplex, is one of the most prevalent and incapacitating manifestations [76] and is less likely to be modified by current therapies [77,78,79]. Most evident are stunted growth and thoracolumbar kyphosis [76,80,81]. Other abnormalities include flattened vertebral bodies, odontoid hypoplasia at cervical vertebra C2, oar-shaped ribs, short and thickened clavicles, bullet-shaped phalanges, a large skull with a thickened calvarium, and J-shaped sella turcica [82]. The underdeveloped odontoid process at C2 predisposes the patients to atlantoaxial instability [83,84,85], which together with spinal stenosis may cause life-threatening spinal cord compression. Dysplastic femoral heads predispose patients to hip osteoarthritis [86,87]. Other manifestations in the lower extremities include coxa valga and genu valgum. Joint symptoms, such as joint stiffness and limited range of joint mobility, stem from GAG accumulation and secondary pathogenic cascades in the ligaments and capsule around the joints [88,89].

Individuals with MPS I often suffer from reduced bone mineral density (BMD) [90,91], which persists despite treatment by hematopoietic cell transplantation (HCT) and/or enzyme replacement therapy (ERT) [90,91,92]. This portends an increased risk for fracture in individuals with MPS I as they age.

The accumulation of partly digested GAGs in the ECM and the loss of function of GAGs contribute to the development of these musculoskeletal deficiencies [88]. GAG binding to cytokines, chemokines, growth factors, adhesion factors, and enzymes generates a gradient of these molecules that controls cell proliferation and migration [93,94] (see also Section 7.2). Disruption of the equilibrium between GAGs and their binding partners impacts proper cell function and cell-to-cell communication. The epiphyseal growth plates in patients affected by MPS I [94,95,96,97] and in animal models of MPS I [98] are disorganized and show large chondrocytes with large vacuolar contents [99]. Accumulation of GAGs at the growth plates [100,101] may inhibit osteoclastic collagenase cathepsin K, subsequently preventing the cleavage of collagen [102] and interfering with endochondral ossification [92]. This likely causes the presence of cartilage in the primary calcification zone, which further disrupts organization of the growth plate [103]. Articular chondrocyte apoptosis appears to be mediated through accumulation of dermatan sulfate (DS). DS is structurally similar to lipopolysaccharide (LPS) and activates the TLR-4 signaling pathway [104] (see also Section 7.2), initiating the release of proinflammatory cytokines, which eventually leads to articular chondrocyte apoptosis [105,106].

### Diagnostic Tests for Skeletal Deficiencies and Joint Symptoms

Imaging techniques for dysostosis multiplex include radiography to appreciate the skeletal alterations. MRI can help to assess the dimensions of the growth plates and is particularly helpful to assess the degree of spinal cord compression, although special care has to be taken when using anesthesia in MPS I patients [76,107]. Bone mineral density (BMD) can be evaluated via dual energy X-ray absorptiometry (DXA) [91] in MPS I patients and in the larger animal models of MPS I. However, DXA may not be the method of choice for BMD analysis in mice, and other methods, such as micro-CT may be better suited for small animal research (Hampe CS in press). Range of motion tests for different joints (fingers, elbows, shoulders, hips, and knees) should be part of the physical exam.

## 5. Symptoms Emerging after 1 Year of Life

### 5.1. Cognitive Impairment

In children with MPS IH, cognitive development slows down beginning around 6–9 months of age, stagnates during the second year of life, and progresses to a decline thereafter, unless the patient receives appropriate treatment at a young age [108]. Even after HCT, children often show below normal cognitive skill levels [79,109,110,111]. Manifestation of CNS involvement besides impaired cognition include poor attention, adaptive skills, and speech/language function [110,112,113]. However, some of the observed impairments, particularly speech and language, may be exacerbated by other disease-specific manifestations, such as an enlarged tongue or hearing loss as described above. Difficulties in development of motor skills may be further affected by disease-specific effects on joint motion, peripheral neuropathy, visual acuity, and musculoskeletal deficiencies.

The pathomechanisms involved in cognitive impairment are not fully understood, but appear to be dependent on defects in heparan sulfate (HS) metabolism, as individuals with MPS disorders that retain normal HS catabolism show normal cognitive development [114,115]. The association of HS with cognitive impairment may suggest an involvement of abnormal cell signaling [116,117]. Notably, HS and its proteoglycans are known to be crucial components in the ECM and contribute to cell signaling [118,119] and abnormal sulfation pattern or degradation products may interfere with neuronal functioning [120]. Accumulation of GAGs in lysosomes also interferes with lysosomal function, which has particular consequences in the CNS. It has been reported that accumulated GAGs bind and inhibit lysosomal hydrolases including ganglioside degrading enzymes [121,122] (see also Section 7.2). Consequently, GM2 and GM3 gangliosides accumulate in the CNS of MPS I patients and animals [123,124]. Indeed, GAG and ganglioside levels in the brain of MPS IH patients can exceed those found in healthy subjects by up to 6-fold [121,123].

Anatomically, MPS IH patients display abnormalities in the white matter [111,114,125,126,127] dilated perivascular spaces (PVS), atrophy [7,128], cervical spinal cord compression, and hydrocephalus [129,130,131]. The latter may be caused by interference with CSF absorption by GAG-laden cells and extracellular GAG deposits. Importantly, the anatomical changes in the brain do not necessary correlate with neurocognitive impairment [7]. Consequent increases in intracranial pressure can be causal for some of the CNS manifestations, including loss of cognitive function that is irreversible even after shunting, and also for ocular problems, as discussed above (see Section 3.2.2).

The animal models of MPS I show similar CNS changes as those observed in patients [56,132,133,134,135,136]. Behavioral and cognitive defects have been observed in MPS I mice [137,138,139,140] but skeletal abnormalities and sensatory deficiencies need to be considered when analyzing the results [141]. No cognitive deficits were found in MPS I dogs [142].

### 5.2. Diagnostic Tests for Cognitive and Adaptive Skills

Exams that target mental age, rather than chronological age, are used to assess cognitive function in MPS I patients. For children who are mentally functioning below age 42 months, the third edition of the Bayley Scales of Infant and Toddler Development (BSID-III) has been recommended, while a variety of tests are available for children who are functioning above this level. Comprehensive reviews of these tests in the evaluation of cognitive function for MPS I patients are provided elsewhere [143,144]. For adaptive behavior, the Vineland Adaptive Behavior Scales Second or Third Editions (VABS-II or III) appear to be well suited to characterize MPS I patients [144].

Presence of specific partially degraded GAGs presents a potentially valuable biomarker for CNS involvement. Measurements of residual IDUA activity and GAG fragments successfully identified MPS I patients with and without CNS involvement [145] and may indicate disease progression [146] and disease severity [147]; a reduction in these fragments has been associated with better cognitive outcomes after treatment. Notably, lower levels of HS nonreducing ends in the CSF of Hurler patients is associated with better neurocognition, and reduction in HS nonreducing ends was observed in patients who responded favorably to HCT [148]. In mice, the levels of HS in brain tissue are useful biomarkers for neurological involvement [149]. However, these tissues are not available in patients and it remains to be seen whether HS levels in the CSF are equally reliable. MRI can be used to determine morphological deficits associated with MPS I [114,150,151].

## 6. Cardiac Manifestations

Cardiac disease is a common complication in patients affected with MPS I and progressive cardiac valve pathology can be found in the majority (60–90%) of patients [11,152,153,154]. Congestive heart failure is a common cause of death in patients with severe MPS I [155]. Age of onset and pace of progression vary considerably [152], and cardiac disease can be seen already in infancy [155,156,157,158]. Cardiac manifestations in MPS I patients involve mainly valvular heart disease and coronary artery narrowing. However, occlusion of the abdominal aorta and renal arteries and associated systemic hypertension has also been described (Figure 4) [159].

### 6.1. Valve Abnormalities

GAGs are normal components of cardiac valves and the great vessels [160,161]. Thus, it is not surprising to find a strong association between MPS I and cardiac valve disease [162,163]. The most commonly affected valves are the mitral and the aortic valves that are significantly thickened [164,165] by progressive infiltration of GAG-laden activated valvular interstitial cells into the valvular tissues [166] and excessive deposits of collagen [167]. Functional outcomes are poor mobility of the valves, regurgitation, and, to a lesser extent, stenosis [168]. Both can lead to atrial and/or ventricular volume overload, ventricular dilatation, ventricular hypertrophy, and ultimately to systolic and diastolic dysfunction [163]. In general, the left-sided valves are more severely affected [154,162,167,169].

### 6.2. Coronary Artery Disease

GAG deposits in the blood vessels also cause narrowing of the coronary arteries [164,167,170]. The accumulation of myointimal collagen further contributes to narrowing of the lumen. This luminal narrowing can be greater than 75% [167]. It remains unclear why the coronary arteries are preferentially affected, while other arteries are not.

### 6.3. Other Vascular Changes

Individuals with MPS I suffer also from wall thickening of the great vessels, resulting in narrowing of the lumen [159]. The reduced aortic elasticity observed in MPS I [171] may be linked to GAG-mediated disruption of elastin fibers, resulting in elastin that is both decreased in content and abnormal in structure [172] (see below Section 7.3).

### 6.4. Diagnostics for Cardiac Manifestations

Valvular disease leading to regurgitation and/or stenosis can be identified by two-dimensional echocardiography and Doppler examination. Coronary angiography may be indicated when a strong suspicion of coronary artery disease is present [152,173,174].

### 6.5. Cardiac Manifestations in Animal Models

All three animal models of MPS I demonstrate some degree of cardiac disease, with aortic and mitral valve thickening and aortic dilation due to infiltration of the valve spongiosa with GAG-laden cells and collagen accumulation [16,56,175,176,177,178]. In addition, the aorta contains enlarged cells in the tunicae intima and media, leading to thickening of the aortic walls [16,56,178]. In MPS I dogs, the coronary artery shows marked expansion of the intima contributing to thickened coronary arteries [16,178]. The hearts of MPS I mice are overall enlarged with thickened septal and posterior walls, whereas the diameter of the inner chambers remains unchanged. MPS I mice have been particularly useful for longitudinal investigations of cardiac manifestations. GAG accumulation in the heart can be noted already at 3 months of age and reaches significantly elevated levels at 7 months of age [177]. Ventricular dysfunctions usually do not present until 6 months of age, followed by reduction of contractile function and ejection function by 10 months of age [179] and significant dilation of the aortic root after the age of 8 months [177,179]. There are notable differences between cardiac manifestations in MPS I patients and mice. Although the mitral valve is the most affected valve in humans, the aortic valve shows most dysfunction in MPS I mice. Similarly, the aortic valve remains mainly intact in MPS I patients, while the mitral valve remains competent in MPS I mice. MPS I mice show relatively mild lesions in their proximal aorta, whereas MPS I patients demonstrate arteriosclerotic coronary or peripheral arterial lesions [176]. Finally, aorta dilation with aortic root dilatation is common in mouse models of MPS I [177] and is likely due to enzymatic degradation of elastin [180,181], whereas this is uncommon in patients with MPS I [152] and only observed in patients with severe Hurler syndrome [182].

## 7. Pathogenic Mechanisms

### 7.1. Cellular Pathology

On the microscopic level, the disease is characterized by the presence of GAG-laden cells in all tissues. These cells consist mainly of macrophages and fibroblasts, but also include bone, muscle cells, and neural cells. Extracellular GAG deposits cause the tissue to absorb more water and become inflated. GAG deposits and GAG-laden cells also interfere with the organization of collagen and elastin fibers.

### 7.2. Inflammatory Immune Responses

Inflammatory responses secondary to GAG accumulation were described in Hurler patients as early as 1960 [183]. In the following subsections, we briefly review how accumulation of GAGs in MPS I may elicit inflammatory immune responses and associated tissue damage.

Sequestering of immune cells in the ECM: The presence of partially degraded HS with abnormal sulfation patterns in the ECM impacts leukocyte and immune cell migration, further exacerbating inflammation [184,185]. Under normal circumstances, HS has a major function in the mediation of cell migration through binding and capture of chemokines [186,187]. The majority of chemokines appear to bind HS [188,189], and their binding preferences are impacted by HS-sulfation patterns. Excessive 2-O-sulfation of HS is observed in MPS I [132] and has been demonstrated to increase binding to CXCL12, thereby sequestering hematopoietic stem cells to the ECM of bone marrow cells and limiting their migration [190].

Activation of the TLR-4 pathway: GAGs are known to activate TLR-4 [191,192,193]. TLR-4 activation results in the increased expression and release of pro-inflammatory cytokines, chemokines, and matrix metalloproteinases [104,105]. TLR-4-mediated TNF-α release is central to the inflammatory immune response observed in MPS I [104], with major impacts on synovial inflammation and joint disease [193,194], neuroinflammation [132,133], cardiovascular disease [195], and chronic pain [196]. Polgreen et al. followed up their initial finding of elevated TNF-α levels in MPS I patients with chronic pain and limited physical function [196] with a small pilot study, where MPS patients received TNF-α inhibitor adalimumab [197]. The treatment resulted in less pain and improved range of motion in the two participating patients with MPS I and MPS II, respectively.

Breakdown of lysosomal membranes: Accumulation of GAGs in the lysosomes may compromise the integrity of lysosomal membranes, although the mechanism by which this occurs is not yet known. The subsequent release of lysosomal cysteine proteases into the cytoplasm [198] may trigger apoptotic signaling pathways, as has been reported for lymphocytes in MPS I mice [199]. Permeabilization of lysosomal membranes also allows for an elevation of pH in the organelles [198] impacting the activity of other lysosomal hydrolases [200,201]. Together, this results in oxidative stress, necrosis, and apoptosis [202].

### 7.3. Biochemical Pathology

The lack of IDUA activity in MPS I results in the accumulation of DS and HS with associated abnormalities in their respective proteoglycans syndecan, glypican, betaglycan, and perlecan (HS), and decorin and biglycan (DS). As a major part of the extracellular matrix (ECM), GAGs and proteoglycans have essential roles in a variety of biological functions, including the proper organization of collagen fibrils and elastin fibers. The incomplete degradation of GAGs leads to impairment of cellular structures and interference with normal ECM functions. Moreover, interruption of GAG recycling results in a shortage of functional GAGs and proteoglycans.

DS-containing proteoglycans function as structural constituents of complex matrices such as cartilage, intervertebral discs, tendons, and corneas. Among their many biological functions, they dictate proper collagen organization and are responsible for corneal transparency. Decorin and biglycan are small leucine-rich proteoglycans whose protein cores carry one or two dermatan/chondroitin sulfate chains on their N-terminus, respectively. Both are present in skin, tendon, cartilage, kidney, cornea, and muscle, while biglycan is also present in bone. Decorin and biglycan have collagen-binding motifs and regulatory functions in collagen fiber growth, size, morphology, and content [203,204]. The binding of proteoglycans to collagen regulates the lateral association of collagen molecules into proper fibrils, and protects collagen fibrils from proteolysis by sterically limiting the access of collagenases to their cleavage sites. The decorin protein core binds noncovalently to an intermolecular cross-linking site near the C-terminus of collagen alpha 1(I) [205], whereas the GAG chains regulate interfiber interactions. Decorin limits the diameter of collagen fibrils [206,207]. Decorin-deficient mice show abnormal collagen fibril morphology in the skin and tail tendons and fragile skin with thinning of the dermis [208]. Other phenotypic characteristics of decorin deficiencies are weak tendons, lower airway resistance, slow wound healing, and delayed angiogenesis.

Biglycan appears to have a prominent role in the regulation of skeletal growth. Biglycan-deficient mice are shorter than their wild-type littermates and show slower growth of long bones and reduced bone mass [209], mirroring the skeletal symptoms in MPS I patients. This phenotype is associated with a marked decline in the number of osteoblasts. Mechanistically, biglycan modulates bone morphogenic protein 4 (BMP-4)-induced osteoblast differentiation [209,210] and blocks BMP-4 activity [211]. Moreover, biglycan affects the Wnt signaling pathway [212].

Both biglycan and decorin regulate collagen fibrillogenesis in the cornea and biglycan-/decorin-deficient mice show abnormal arrangements of collagen fibrils in their corneas with clouding, similar to that associated with MPS I [213].

HS proteoglycans are associated with the cell surface or the pericellular matrix. Syndecans are transmembrane proteoglycans carrying both HS and chondroitin sulfate chains at their N-terminal ectodomain. Syndecan-1 is predominantly expressed on the basolateral surface of epithelial cells [214] and interacts with multiple ECM components, including collagen and fibronectin via its extracellular GAG units, whereas the protein core interacts with integrin. Syndecan thereby bridges the intracellular cytoskeleton to the ECM [215]. Moreover, syndecan regulates the biological activity of ligands and acts as a coreceptor to catalyze the interaction between ligand and receptor [216]. Syndecan-1-deficient mice show no major pathologies [217] but are more susceptible for inflammation [218], underscoring regulatory role of syndecan in the immune response (see also Section 7.2).

Perlecan is based on a ~500-kDa protein core located at the apical cell surface [219] and basement membranes [220]. HS chains are located at the N-terminus of the protein core, where they interact with growth factors, growth factor receptors, collagen, and other ECM proteins [221]. In the basement membrane, perlecan binds and links collagen IV, nidogen, and laminin in order to further strengthen the basement lamina [222]. Perlecan regulates various biological processes. Most relevant to MPS I are the development of blood vessels and cartilage [223] and endochondral bone formation [224]. Perlecan levels are significantly reduced in patients with Schwartz–Jampel syndrome, which presents with short stature, kyphosis, skeletal dysplasia, and myotonia [225], thereby displaying some of the musculoskeletal symptoms seen in MPS I patients. Perlecan-deficient mice show poor bone quality [226].

### 7.4. Biochemical Mechanism of Loss of Cardiac Elasticity

The pathogenic mechanism underlying the loss of elasticity in MPS I hearts has been studied in detail using the mouse model. Elastin is the main component of elastic fibers essential for the arterial elasticity and is found at high concentrations in the arterial and aortic walls and the ventricles. Elastin fragmentation has been observed in the aorta of MPS I mice [176,227], ascending aorta and heart valves of MPS I dogs [16], and valves and coronary arteries and aortic walls of MPS I patients [167,177]. This defect could be caused either by insufficient assembly [172] or by increased activity of elastin proteases [227]. The observation that elastin fibers are normal in young MPS I mice suggests that the initial elastin synthesis is unaffected and that the defect lays in the activated degradation of elastin fibers [227]. Moreover, elevated elastolytic proteases MMP-12 and cathepsin S correlated with progressive elastin fragmentation and aortic dilatation in 6 month-old mice [228,229]. Abnormal arrangements of collagen can be observed in the cardiac valves of MPS I mice and in cardiac valves and cardiac arteries of patients [167,177]. As discussed above for elastin, dysregulated proteolytic activity may be the cause for this abnormality. Indeed, cathepsin activity in MPS I mice is 10-fold increased over that found in normal mice [179]. Cathepsin B (CtsB) is a lysosomal enzyme found at high levels in activated macrophages. Baldo et al. hypothesized that the accumulation of undegraded GAGs attracts the infiltration and activation of macrophages [179]. Although CtsB is usually localized in the lysosomes, MPS I-mediated compromise of the lysosomes may allow leakage of the enzyme. Indeed, CtsB levels are increased in MPS I mouse serum [230], and inhibition of CtsB caused partial improvements in both aorta and heart valves [229,230].

## 8. Conclusions

The early diagnosis of MPS I is crucial to prevent the development of potentially fatal complications. This review summarizes the major disease manifestations in the order in which they typically develop and diagnostic methods for their assessment. The cellular and biochemical pathomechanisms of MPS I are discussed to aid in better understanding the disease.

## Figures and Tables

**Figure 1 cells-09-01838-f001:**
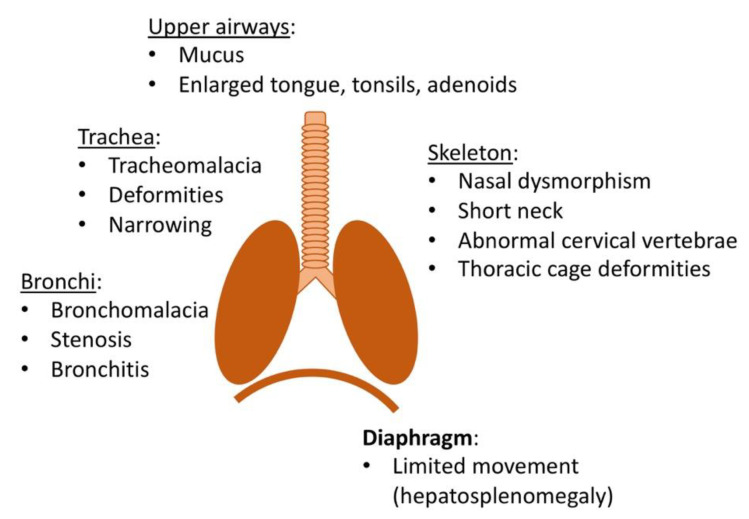
Overview of manifestations affecting respiratory function in mucopolysaccharidosis type I (MPS I).

**Figure 2 cells-09-01838-f002:**
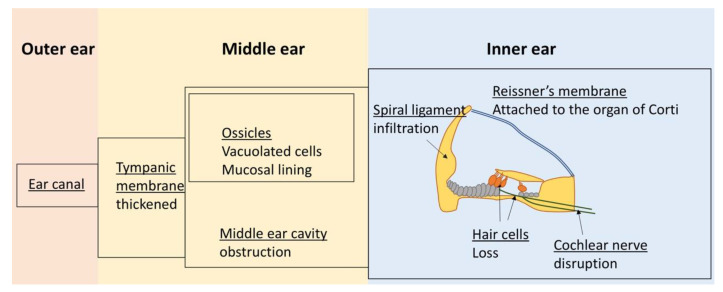
Overview of manifestations affecting auditory function in MPS I.

**Figure 3 cells-09-01838-f003:**
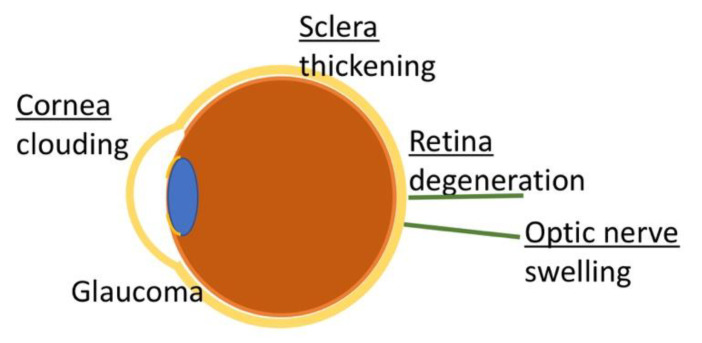
Overview of ocular manifestations in MPS I.

**Figure 4 cells-09-01838-f004:**
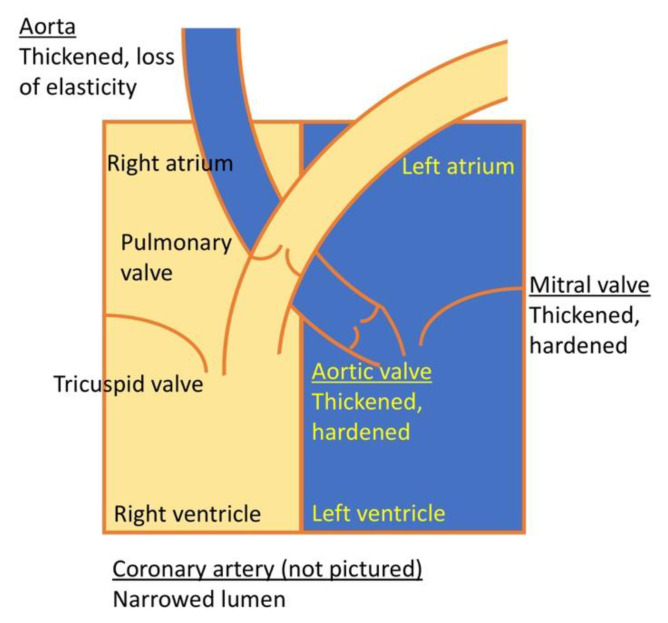
Schematic overview of manifestations affecting cardiac function in MPS I.

**Table 1 cells-09-01838-t001:** Natural history of symptoms in severe mucopolysaccharidosis type I (MPS I).

0–6 Months	6–12 Months	Late
Recurrent rhinitis	Hearing loss	Progressive cognitive slowing, then loss
Upper airway obstruction	Lower airway obstruction	
Coarse facial features	Visual impairment	
Thoracolumbar kyphosis	Further musculoskeletal defects	
Hernias	Delayed motor milestones	
Hepatosplenomegaly		
Cardiovascular defects		

**Table 2 cells-09-01838-t002:** Auditory manifestations in MPS I.

	Human	Mouse	Dog
Hearing loss	Within the first year of age	Within the first year of age	Yes
**Middle ear**			
Partial obstruction of middle ear cavity	Yes	NA	Yes
Presence of GAG-laden cells	Yes	NA	Yes
Thickened tympanic membranes	Yes	NA	Yes
Ossicles covered by GAG-positive mucosal lining, infiltrated by GAG-laden cells	Yes	NA	Yes
Otitis media	Yes	Yes	No
**Inner ear**			
Otitis interna	Yes	Yes	No
Infiltration by GAG-laden cells	Yes		Yes, but at older age
Degeneration of the organ of Corti	Yes		No
Loss of cochlear hair cells	Yes	Yes	No
Damage to cochlear nerve	Yes		Yes
Damage to cochlear fibrocytes		Yes	Yes

NA: not analyzed.

**Table 3 cells-09-01838-t003:** Diagnostic auditory exams.

Exam	Main Targeted Area
Audiometry	Overall hearing ability
Bone conduction testing	Inner ear
Tympanometry	Middle ear
ART	Middle ear
BAEP	Inner ear
OAE	Inner ear (outer hair cells)
Static acoustic impedance	Middle ear (eardrum)

**Table 4 cells-09-01838-t004:** Ocular manifestations in MPS I.

	Human	Assessment	Animal Model
**Corneal Clouding**	Yes	Slit lamp exam	Yes (all models)
**Optic Nerve Swelling**	Yes	Pupil reaction to light, visual field evaluation, fundus evaluation	
**Retinal Degeneration**	Yes (late symptom)	Color vision test	Mice and cats
**Glaucoma**	Yes (~4% of patients)	Measurement of IOP	

**Table 5 cells-09-01838-t005:** Diagnostic tests for ocular manifestations.

Ocular Manifestation	Test
**Refractive errors**	Visual acuity test
**Strabismus**	Stereopsis assessment
**Corneal clouding**	Slit lamp exam Iris Camera in vivo confocal microscopy
**Glaucoma**	Visual field exam IOP Optical coherence tomography (OCT)
**Retinopathy**	Complete fundus exam OCT Ultrasound (A- and B-scan) Electroretinography
**Optic nerve damage**	OCT Visual field exam

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
