# Peer review of "Mucopolysaccharidosis Type I: A Review of the Natural History and Molecular Pathology"

_cells, 2020, doi:10.3390/cells9081838_

Round 1

Reviewer 1 Report

The authors have presented interesting review paper “Mucopolysaccharidosis Type I: A Review of the natural history and molecular pathology” for Cells. The author summarizes recent findings on the pathogenesis of mucopolysaccharidosis type I very well. There are some questions.

1 The author describes in detail the natural history of mucopolysaccharidosis type I, but there is no section on bone deformity and joint symptoms.

Please describe in detail by providing a section on bone deformity and joint symptoms.

ï¼’Although the author has a biochemical description in the part that describes the pathology (line 384), it does not describe tissue deformation and functional abnormality due to inflammation associated with GAGs accumulation. Please describe them additionally.

3 On line 52, only “at” is in bold.

Author Response

Reviewer 1 The authors have presented interesting review paper “Mucopolysaccharidosis Type I: A Review of the natural history and molecular pathology” for Cells. The author summarizes recent findings on the pathogenesis of mucopolysaccharidosis type I very well. There are some questions.

We greatly appreciate the reviewer’s kind assessment of our manuscript. A point-by-point response to the comments are provided below.

1 The author describes in detail the natural history of mucopolysaccharidosis type I, but there is no section on bone deformity and joint symptoms. Please describe in detail by providing a section on bone deformity and joint symptoms.

We now added a new section describing bone deformity and joint symptoms (pages 9-10, lines 277-318).

ï¼’Although the author has a biochemical description in the part that describes the pathology (line 384), it does not describe tissue deformation and functional abnormality due to inflammation associated with GAGs accumulation. Please describe them additionally.

We added a new section entitled “6.b Inflammatory immune responses.” summarizing pathways leading to inflammation and the role of inflammation in neuroinflammation, point disease, cardiovascular disease and chronic pain (pages 12-27, lines 431-458).

3 On line 52, only “at” is in bold.

This has been corrected.

Reviewer 2 Report

In this review, Hampe and coauthors describe in detail the clinical features of MPS I patients expanding their study from human patients to animal models. It is of interest and updated from the big literature in the field how they correlate each clinical phenotype of the MPS I patients with the diagnostic tools used in clinical practice and with the pathogenetic mechanisms causing the broad spectrum of diseases.

Authors are experts in the field and the review is presented in a novel shape compared to the recent reviews on the MPS I. The review is very interesting and well structured.

I have some minor comments that authors should amend:

1) There are few grammatical errors. Please check English spelling all over the text before publication.

2) Please correct punctation pg. 2 ln. 56-57.

3) Table 1 should be placed immediately after it is cited in the text for the first time and not after 2 pages.

4) Figure 1 should be placed immediately after it is cited in the text for the first time and not before.

5) Pg. 7 ln. 231, please correct title 3.b2 as “Optic nerve swelling”.

6) What are the dots in Table 5. Please eliminate them.

7) Pg. 9 ln. 292, please add two new important references in the field after “abnormal cell signaling”:

  1. De Pasquale, V.; Pavone, L.M. Heparan sulfate proteoglycans: The sweet side of development turns sour in mucopolysaccharidoses. Biochim. Biophys. Acta 2019, 1865, 165539,doi:10.1016/j.bbadis.2019.165539.
  2. Fecarotta, S.; Tarallo, A.; Damiano, C.; Minopoli, N.; Parenti, G. Pathogenesis of mucopolysaccharidoses, an update. Int. J. Mol. Sci. 2020, 21, doi:10.3390/ijms21072515.

8) Move Figure 4 before 5.a Valve abnormalities.

9) Pg. 13 ln. 454, and ln. 462-463, please add a new important reference in the field of Cathepsins in MPS I after “6 month-old mice” and after “aorta and heart valves”:

  1. De Pasquale, V.; Moles, A.; Pavone, L.M. Cathepsins in the pathophysiology of mucopolysaccharidoses: New perspectives for therapy. Cells 2020, 9, doi:10.3390/cells9040979.

Author Response

Reviewer 2:

Authors are experts in the field and the review is presented in a novel shape compared to the recent reviews on the MPS I. The review is very interesting and well structured.

We thank the reviewer for his/her careful evaluation of our manuscript and the excellent suggestions. A point-by-point response to the comments is provided below.

I have some minor comments that authors should amend:

  • There are few grammatical errors. Please check English spelling all over the text before publication.

A number of native English speakers have checked the amended manuscript.

2) Please correct punctation pg. 2 ln. 56-57.

This has been amended.

3) Table 1 should be placed immediately after it is cited in the text for the first time and not after 2 pages.

Table 1 has been moved accordingly (page 2, line 69).

4) Figure 1 should be placed immediately after it is cited in the text for the first time and not before.

Figure 1 has been moved accordingly (page 3, line 104).

5) Pg. 7 ln. 231, please correct title 3.b2 as “Optic nerve swelling”.

This has been changed.

6) What are the dots in Table 5. Please eliminate them.

Bullet points have been removed (pages 8-9, line 276).

7) Pg. 9 ln. 292, please add two new important references in the field after “abnormal cell signaling”:

  1. De Pasquale, V.; Pavone, L.M. Heparan sulfate proteoglycans: The sweet side of development turns sour in mucopolysaccharidoses. Biochim. Biophys. Acta 2019, 1865, 165539,doi:10.1016/j.bbadis.2019.165539.
  2. Fecarotta, S.; Tarallo, A.; Damiano, C.; Minopoli, N.; Parenti, G. Pathogenesis of mucopolysaccharidoses, an update. Int. J. Mol. Sci. 2020, 21, doi:10.3390/ijms21072515.

Both references have been added.

8) Move Figure 4 before 5.a Valve abnormalities.

Figure 4 has been moved accordingly (page 11, line 378).

9) Pg. 13 ln. 454, and ln. 462-463, please add a new important reference in the field of Cathepsins in MPS I after “6 month-old mice” and after “aorta and heart valves”:

  1. De Pasquale, V.; Moles, A.; Pavone, L.M. Cathepsins in the pathophysiology of mucopolysaccharidoses: New perspectives for therapy. Cells 2020, 9, doi:10.3390/cells9040979.

The reference has been added.

Reviewer 3 Report

Manuscript ID: cells-885016

Type of manuscript: Review

Title: Mucopolysaccharidosis Type I: A Review of the natural history and molecular pathology

The authors reviewed the natural history and molecular pathology of mucopolysaccharidosis type I. They discussed common phenotypic manifestations of the disease, in the order in which they typically develop. Key manifestations were described in detail.

This article is well-written and important. There are only some minor points needed to be revised.

  1. Page 1, line 18: “α-l-iduronidase” should be changed to “α-L-iduronidase”.
  2. Page 5, line 181: “Manifestations” should be changed to “manifestations”.
  3. Page 7, line 232: “nerve” should be changed to “Nerve”.
  4. Page 8, line 273: “Manifestations” should be changed to “manifestations”.
  5. Page 15, line 560: The information of Reference 38 is incomplete. Please check.

Author Response

Reviewer 3

We thank the reviewer for the careful comments and have amended the manuscript accordingly.

  1. Page 1, line 18: “α-l-iduronidase” should be changed to “α-L-iduronidase”.
  2. Page 5, line 181: “Manifestations” should be changed to “manifestations”.
  3. Page 7, line 232: “nerve” should be changed to “Nerve”.
  4. Page 8, line 273: “Manifestations” should be changed to “manifestations”.
  5. Page 15, line 560: The information of Reference 38 is incomplete. Please check.

The manuscript has been changed accordingly.